# Impact of Covid -19 incidence rate and government-initiated risk communication measures on individual's NPI practices

**Yifokire Tefera** *, **Abera Kumie, Damen Hailemariam, Samson Wakuma, Teferi Abegaz, Mulugeta Tamire, Shibabaw Yirsaw**

Department of Preventive Medicine, School of Public Health, College of Health Sciences, Addis Ababa University, Addis Ababa, Ethiopia

* yifoomitu@yahoo.com

## Abstract

### Background

Non-pharmaceutical interventions (NPI) are the most widely recognized public health measures recognized globally to prevent the spread of Covid-19. NPIs' effectiveness may depend on the type, combination of applied interventions, and the level of proper public compliance with the NPIs. The expected outcome of behavioural practices varies relative to the intervention duration.

### Objectives

This study aimed to assess the trend of community compliance to NPI with Covid-19 incidence and government-initiated interventions, and its variation by residence and sociodemographic characteristics of people.

### Methods

A weekly non-participatory field survey on individuals' NPI practices was observed from the 41st epidemiological week of October 5th, 2020, to the 26th epidemiological week of July 4th, 2021, a total of 39 weeks. The survey covered all 14 regional and national capital cities in Ethiopia. Data collection for the three NPI behaviours (i.e., respiratory hygiene, hand hygiene, and physical distance) was managed weekly at eight public service locations using the Open Data Kit (ODK) tool. The Covid– 19 incidence data and public health measures information from August 3rd, 2020 to July 4th, 2021 were obtained from the Ethiopian Public Health Institute (EPHI).

### Results

More than 180,000 individuals were observed for their NPI practice, with an average of 5,000 observations in a week. About 43% of the observations were made in Addis Ababa, 56% were male and 75% were middle age group (18–50 years). The overall level of NPI compliance was high at the beginning of the observation then peaked around the 13th– 15th

**Data Availability Statement:** In the manuscript and additional SPSS data file uploaded

**Funding:** The author(s) received no specific funding for this work.

**Competing interests:** The authors have declared that no competing interests exist.

epidemiological weeks then declined during the rest of the weeks. The peak NPI compliance periods followed the high Covid-19 death incidence and government-initiated intensive public health measures weeks. Respiratory hygiene had the highest compliance above 41% whereas hand hygiene was the lowest (4%). There was a significant difference between residents of the capital city and regional cities in their level of compliance with NPI. Females comply more than males, and individuals had increased NPI compliance at the bank service and workplaces compared to those in the transport services at $P = 0.000$.

## Conclusion

An increased level of compliance with NPI was observed following intensive government-initiated Covid-19 prevention measures and an increased Covid-19 death incidence. Therefore, the intensity of government-initiated risk communication and public advocacy programs should be strengthened, possibly for similar respiratory disease pandemics in the future.

## Introduction

Individual-level Non-Pharmaceutical Interventions (NPIs) are part of the public health and social measures (PHSM) of the World Health Organization (WHO) to prevent the spread of Covid-19 [1]. In the early stages of Covid-19, governments from all over the world started a variety of PHSMs that can be implemented by individuals, institutions, communities, and local and national government bodies due to the lack of clear and proven medical treatments for Covid-19 and the paucity of scientific evidence, globally [1, 3–5]. Governments were under pressure as disease incidence rose. In the current study, NPIs are specifically focused on an individual's behaviour of mask use, physical distancing, and hand hygiene [2].

Individual behavioral practices are advised as vital to halt the development of Covid-19, although countries have employed a variety of context specific, diverse PHSMs [6]. Few studies attempted to assess the efficacy of various NPI practices adopted by governments in different countries [3, 4]. Haug and colleagues [3] gathered thousands of implemented NPIs from 79 different countries and discovered that there is no one best NPI that can stop the spread of Covid-19. Instead, researchers found several combinations of interventions that can drastically reduce the transmission rate. Curfews, lockdowns, as well as restricting locations for public gatherings, were the most effective NPIs in their study [3]. A comparable cross-country study assessed the efficacy of eight NPI related to case identification, environmental measures, health care, public health capacity, resource allocation, risk communication, social distancing, travel restriction, and returning to normal life. The study demonstrated that risk communication had the greatest impact on the population. The authors emphasized that risk communication strategies such as providing general information about Covid-19 or using a face mask, were less likely to enforce any particular behavior on individuals but had a significant impact on the public [4].

WHO created "Covid-19 Global Risk Communication and Community Engagement Strategy" after considering risk communication to be a potent tool to alter peoples' behavior and willingness to adhere to public health measures [7]. WHO, in collaboration with national authorities, institutions and researchers, continues to monitor the public health events associated with Covid 19. Furthermore, WHO documented the different measures taken by the countries in their profile [8]. The major action categories include biological, individual,

international travel measures, social and physical distance, surveillance and response measures and other measures. In Ethiopia, about 59 different government-initiatives for Covid-19 prevention and control were recorded during the study period [9]. Comparison of the impact of interventions was limited since many NPIs were introduced at the same time or at separate times [3, 6, 10]. Sharma and colleagues highlighted the importance of local context and suggested a dozen assumptions be taken into account for a reliable estimation of the effect of NPIs [5, 10]. However, several studies concurred and suggested that combined NPIs were the most efficient way to limit the spread of Covid-19 [3–6, 11].

In Ethiopia, the first Covid-19 case was found in the second week of March 2020, and at the end of the current study week, a total number of reported cases and deaths reached to 275,435 and 4,331, respectively [12, 13]. Since the recognition of the case with Covid-19 infection, the Federal Ministry of Health's (FMoH) Ethiopian Public Health Institute (EPHI) established a National Public Health Emergency Operation Center (PHEOC) to lead the national coordination in collaboration with regional governments and global partners such as WHO. Each regional government has its own Public Health Institute that collaboratively work with the federal EPHI. The national PHEOC collect data of regional Covid-19 cases and deaths daily [2, 14]. Starting from the 24[th] WHO Epidemiological week of 2020, PHEOC published a weekly summary of new infections, recoveries and fatalities at national level. Two peaks were observed for new infections and new deaths, around the 34[th] and 13[th] epidemiological weeks of the year 2020 and 2021, respectively (S1 Raw image) [12].

The federal government of Ethiopia had put several PHSMs into effect across the nation. Although several campaigns and initiatives implemented against Covid-19, the two most critical milestones in Ethiopia for Covid-19 prevention measures were the declaration of State of Emergency issued on the 20[th] of April 2020 and the Covid-19 Pandemic Prevention and Control Directive No 30/2020. Although, the directive contains implementation details but a full enforcement campaign of the regulations was launched on the last week of March 2021 [15–17]. In addition, in the first week of February 2021 the 'No mask No Service' campaign was also launched with the provision of a state of emergency. Some of the PHSMs include closing of international land border, cancellation, school closure, and restriction of public gatherings, public awareness campaigns, enforcement to use personal protective measures, case detection, isolation, and quarantine, among others. Regional governments had also placed various implementation strategies within the local context [1, 14]. Since Ethiopia is known by its wide range of sociocultural diversity, the commitment, technical and enforcement capabilities to adopt federal policies and guidelines are varied across the regions. Furthermore, institutions and business facilities (e.g., religious places, schools, banks and market places) implement the national guidelines at different level of capacity.

The three NPI measures (i.e., mask use, physical separation, and hand hygiene) were the most frequently used Covid-19 interventions at individual basis. In the second week of June 2020, Deressa, et al investigated these personal safety precautions among Addis Ababa city government employees. They found that more than 90% of the participants used face masks, washed their hands, and maintained physical distancing at the workplace [18]. A population of 12,056 residents of Addis Ababa city participated in a weekly NPI monitoring through a non participant observation from April to June 2020 for 10 uninterrupted weeks. The study found an increase in proper hand hygiene from the baseline of 24% to 33%, better physical distance from 34% to 43%, and mask use from 24% to 77% in week 10[th] [2]. The study showed that it took 6 weeks to reach the peak of practicing proper respiratory hygiene after the baseline [16]. Another study in the country, at Bule Hora Town, found that, during the last weeks of September 2020 (38%) of the participants had good social distancing practices [19].

Most studies looked for the impact of NPIs on the reduction rate of new covid-19 cases. However, knowledge regarding the relation between the rate of new cases and fatalities with the NPI practice was limited. The incidence rate of Covid-19 in a population showed a sporadic trend. The increased rate of transmission and fatalities might trigger the widely use of NPI in a community. We developed a research question to monitor the trends of NPIs after the State of Emergency lifted. The current study focuses on determining the trend of the weekly changes in the three individual-level NPI behaviors and its relationship with the incidence rate and government-initiated measures. This is an extension of the weekly NPI monitoring of non-participatory observational research, with extended period on urban population [2].

## Methods

### Study design, area, setting, and period

A repeated cross-sectional observation tracked the weekly NPI behaviours of people in several public places throughout the urban centers of Ethiopia. The NPI monitoring techniques were first implemented in Addis Ababa, the capital city of Ethiopia, in April 2020, and then covered the entire nation, encompassing all regional capital cities starting from October 2020. Before 2020, there were nine regional states, but the Federal Government of Ethiopia now has 12 regional states and 2 chartered cities administrations. The two new regions, Sidama and Southwest Ethiopia regions were established in June 2020 and the other new region Central Ethiopia Regional State in 2023. This NPI monitoring study is designed and lead by Addis Ababa University, School of Public Health and collaborated with other universities situated in each region. Whereas, the national Covid-19 incidence monitoring data were obtained from the Ethiopian Public Health Institute [13]. Weekly NPI monitoring data from 15 cities, including Addis Abeba, Bahir Dar, Gondar, Adama, Hawasa, Asosa, Gambella, Diredewa, Harar, Hosana, Jigjiga, Jimma, Semera, Mekele, and Welayta Sodo, were obtained for the current study.

Ethiopia's overall population is estimated to be 105,166 000 in 2022, according to CSA projections [20]. The NPI data collection period spanned 39 weeks, from October 4, 2020, to July 4, 2021, for a 2-week interruption during the epidemiological weeks 13[th] and 14[th] of 2021 due to a challenge to avail logistics. The Covid–19 incidence data and government-initiated public health measures information from August 3[rd], 2020 to July 4[th], 2021 were obtained from the Ethiopian Public Health Institute (EPHI).

The three NPI behaviours (mask use, hand hygiene, and physical distance) were observed weekly at eight public service locations, including places of worship, medical institutions, markets, banks, public transportation hubs, restaurants, and workplaces. These sites assumed an increase in COVID-19 transmission when individuals were taking public services, such as marketing, accessing transport services, and attending churches.

### Source and study population

About 23,880,000 Ethiopians were urban inhabitants [20]. Around 7,333,908 people resided in the study's sample cities in total. The location of each city is indicated in Fig 1. The source population for the study was all individuals visiting the selected sites or service facilities during the day of data collection. Study participants were individuals who visited the selected sites during the time of data collection.

### Monitoring protocol, data collection tools, and data collection procedure

This study is an extension of the Covid-19 NPI monitoring in Ethiopia, hence monitoring protocol, data collection tools, and data collection procedure were similar to the description

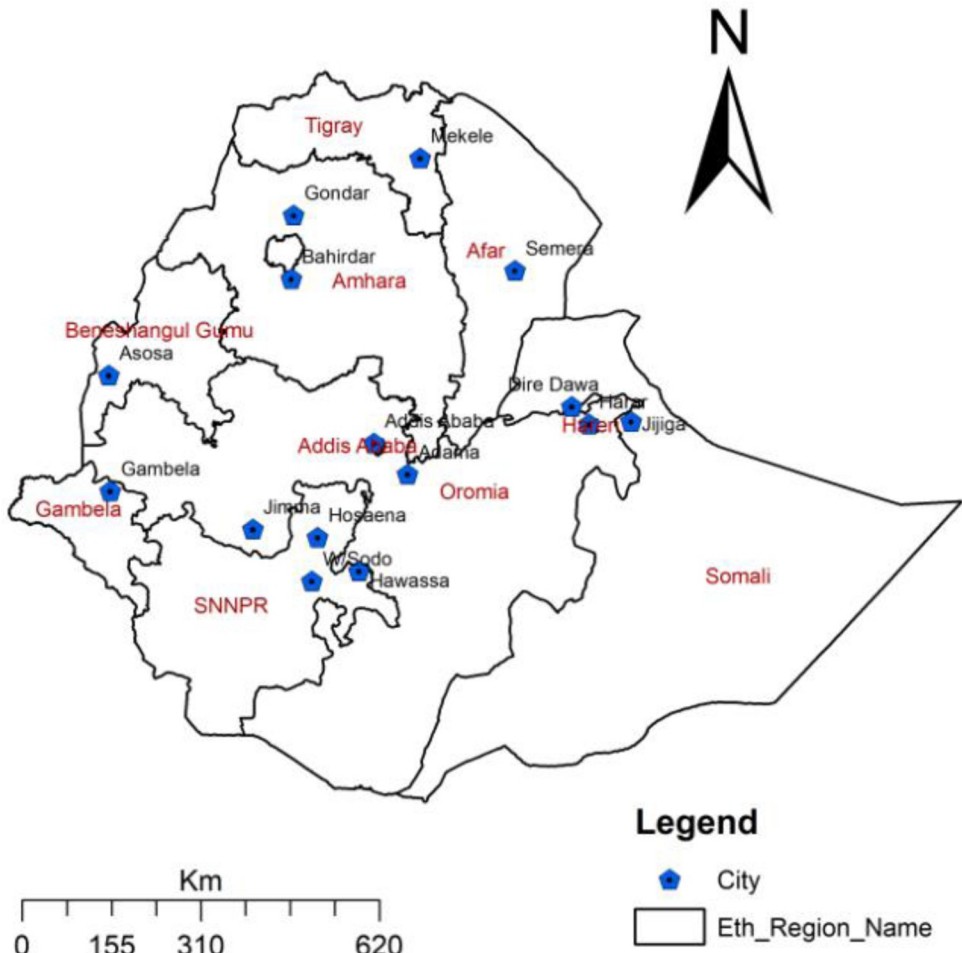

**Fig 1. Location of COVID-19 NPIs monitoring sites by city, Ethiopia, October/2020-July 2021.** Source of the map: Ethiopian Statistical Agency.

mentioned in the previously published report [2]. The monitoring protocol has been approved by the research committee to serve as a standard operative procedure (SOP) for participant selection, operational definition of the NPI practices, time of data collection, the list of public service facilities, etc. The observation of an individual NPI practice at the public service site was anonymous to get a valid behavior of the person. The selection of the person for observation followed a random fashion while seeking service from the facility. Observation start time at each service facility was decided based on pilot work. So, data collectors start the observation who appear in the service facility until the daily required number of observations fulfilled (S1 File). Ethical approval was granted for this purpose. The research assistant collected observation data using a mobile phone and uploaded ODK without being identified by the person (S2 File). This helps to distract suspects in his work, hence maintaining neutrality and data validity. The observation was conducted two days a week, one weekday and one weekend day. However, due to budget limitations, observation was done once a week after the 4th epidemiological week of 2021. Besides, the observation was interrupted during the epidemiological weeks of 14 and 15 due to the lack of avail logistics for field observation during this peak transmission time. Although observation data was collected from nine service facilities, observation from school was started at the epidemiological week 51.

## Data management and analysis

The electronic raw data collected by ODK was immediately channelled to a local server in Addis Ababa University, which was accessed by a data manager who was employed for this purpose. The raw data was obtained in Excel format and was exported to SPSS. All data management, cleaning, coding, recording and analysis were performed on SPSS V 26. All regional city observation data were combined to compare with the capital Addis Ababa city because the city's population made up 51% of the study's total population. We had 39 weeks of monitoring, which involved over 180000 individual observations. On average 5000 observations per week; about 43% of the observations were from Addis Ababa and 47% of the observations were from the 14 regional cities.

The graphs presented in Figs 2 and 3 are the trend of practices over the epidemiological weeks calculated from the proportion of proper NPI users from the total observed population in the week.

The proportion of practices between Addis Ababa Vs regional cities was compared by using the average proportions over the entire 39 weeks using a chi-square test, assuming the sample size is large enough for the power of the test.

Tables, figures, and descriptive statistics were used to present the data. Line and bar graphs were used to display the weekly trend, variations by service facilities, sex, and age group, and appropriate mask use, appropriate hand hygiene, and appropriate physical distancing practices. Proper hand hygiene means either proper hand washing or proper hand sanitizing defined in the protocol (S2 File).

## Ethical considerations

This study was granted ethical approval from the Institutional Review Board (IRB) of the College of Health Sciences at Addis Ababa University. We had permission from IRB not to have consent as the results of the observations were meant to benefit the public at large and we did not seek any identifying information or photos of any individuals. This is indicated in the S3 File. The survey used a random observation tool that does not override the rights of any

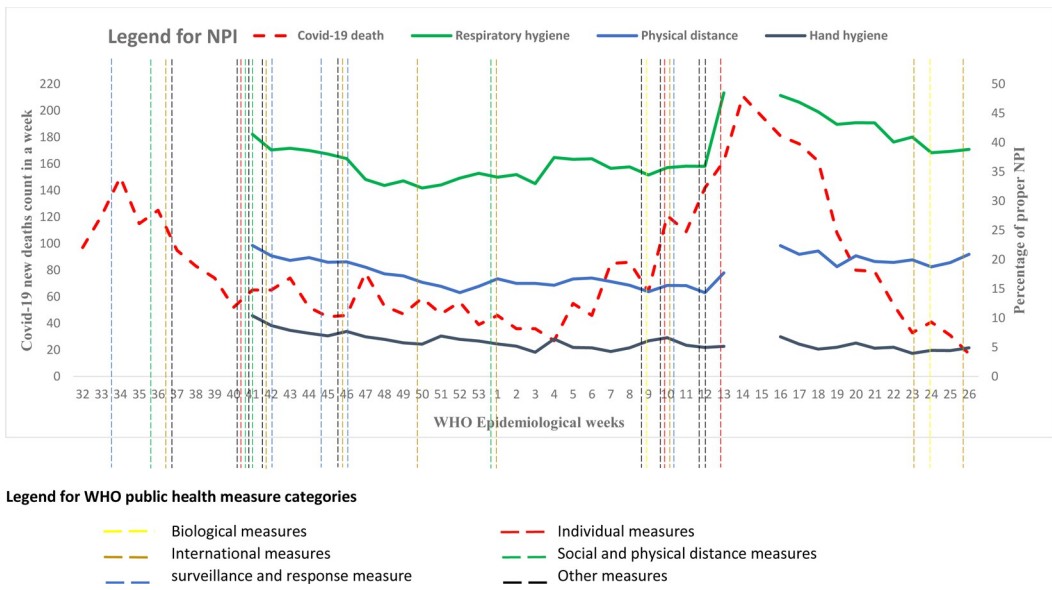

**Fig 2. Trends of proper NPI practices for 39 weeks, covid-19 deaths for 48 weeks and public health measures.**

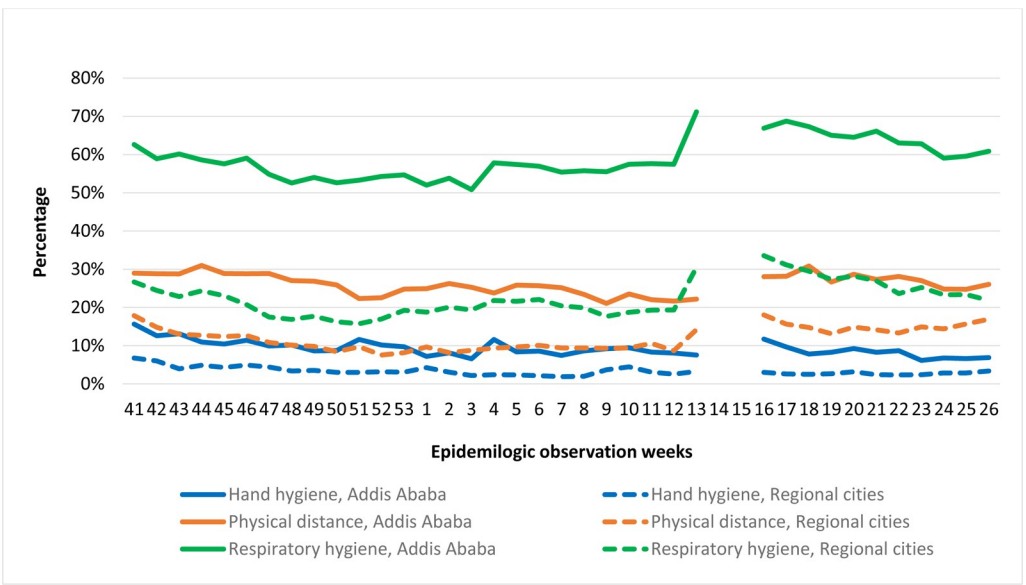

**Fig 3. Trends of proper NPI practice in the Addis Ababa vs regional cities, Oct 5/2020-July 4/2021.**

individual or institution, and the observation was done purely anonymous. Data collection was anonymous and the observers acted similarly to study participants in observation sites when collecting data. A weekly report on the progress of NPI monitoring data was submitted to the Ministry of Health of Ethiopia to take proactive measures.

# Results

## Study participants

More than 180,000 people were observed for their NPI practice for a period of 39 weeks, although the observation was halted for two weeks. On average 5,000 weekly observations have been made from Addis Ababa and regional cities. The proportion of observation from the capital city, Addis Ababa was 43% and from regional cities 57%. Males' participation in the observation had a slightly higher proportion, 56%, compared to females, 44%. The middle age group, 18–50 years were the biggest proportion accounting for 75% followed by the highest age group above 50 years accounts 14% (Table 1).

## Overall trends of proper NPI practice

**Proper respiratory hygiene.** Of the three NPIs, the community had better compliance with respiratory hygiene followed by physical distance than hand hygiene. The respiratory hygiene compliance at national level was 41% at the start of the observation week and declined to the lowest proportion of 32%, and holding 32% - 35% for ten weeks. Then showed weekly progress more than 48% through the epidemiological weeks of 13th– 15th. Subsequently, the proportion of proper respiratory hygiene compliance showed a continuous decline to 39% until the last observation week (Fig 2). However, there was a big difference in respiratory hygiene compliance between Addis Ababa city and major regional cities; there was higher compliance in Addis Ababa city across the observation period than in the regional cities and this difference is statistically significant at $P = 0.000$. Increased compliance was also observed in both Addis Ababa and regional cities around the 12th through the 15th weeks (Fig 3).

**Table 1. Participants observed for NPI in Ethiopia (October 2020 –July 2021).**

| WHO Epi week * | Observation dates | Observation places, n (%) | | Sex, n (%) | Age in years, n (%) | | |
|---|---|---|---|---|---|---|---|
| | | Addis Ababa | Regional cities | Male | < 18 | 18–50 | >50 |
| 41 | Oct 5-11/20 | 2,560 (41%) | 3,674 (59%) | 3,556 (57%) | 436 (7%) | 4,975 (80%) | 823 (13%) |
| 42 | Oct 12-18/20 | 2,580 (42%) | 3,630 (58%) | 3,484 (56%) | 398 (6%) | 4,955 (80%) | 857 (14%) |
| 43 | Oct 19-25/20 | 2,600 (43%) | 3,380 (57%) | 3,369 (56%) | 304 (5%) | 4,806 (80%) | 870 (15%) |
| 44 | Oct 26-Nov 2/20 | 2,600 (42%) | 3,630 (58%) | 3,516 (56%) | 321 (5%) | 5,010 (80%) | 899 (14%) |
| 45 | Nov 3-9/20 | 2,600 (43%) | 3,380 (57%) | 3,389 (57%) | 307 (5%) | 4,820 (81%) | 853 (14%) |
| 46 | Nov 9-15/20 | 2,550 (43%) | 3,370 (57%) | 3,328 (56%) | 314 (5%) | 4,686 (79%) | 920 (16%) |
| 47 | Nov 16-22/20 | 2,600 (43%) | 3,380 (57%) | 3,363 (56%) | 336 (6%) | 4,716 (79%) | 928 (16%) |
| 48 | Nov 23-29/20 | 2,600 (44%) | 3,280 (56%) | 3,277 (56%) | 320 (5%) | 4,649 (79%) | 911 (15%) |
| 49 | Nov 30-Dec 6/20 | 2,600 (43%) | 3,380 (57%) | 3,369 (56%) | 349 (6%) | 4,708 (79%) | 923 (15%) |
| 50 | Dec 7-13/20 | 2,600 (44%) | 3,310 (56%) | 3,319 (56%) | 343 (6%) | 4,641 (79%) | 926 (16%) |
| 51 | Dec 14-20/20 | 2,800 (45%) | 3,370 (55%) | 3,414 (55%) | 529 (9%) | 4,767 (77%) | 874 (14%) |
| 52 | Dec 21-27/20 | 2,800 (45%) | 3,360 (55%) | 3,470 (56%) | 546 (9%) | 4,696 (76%) | 918 (15%) |
| 53 | Dec 28-Jan 3/21 | 2,800 (44%) | 3,600 (56%) | 3,475 (54%) | 724 (11%) | 4,761 (74%) | 915 (14%) |
| 1 | Jan 4-10/21 | 2,790 (46%) | 3,260 (54%) | 3,290 (54%) | 666 (11%) | 4,418 (73%) | 966 (16%) |
| 2 | Jan 11-17/21 | 2,800 (43%) | 3,730 (57%) | 3,597 (55%) | 728 (11%) | 4,876 (75%) | 926 (14%) |
| 3 | Jan 18-24/21 | 2,710 (43%) | 3,550 (57%) | 3,468 (55%) | 670 (11%) | 4,711 (75%) | 879 (14%) |
| 4 | Jan 25-31/21 | 1,800 (43%) | 2,340 (57%) | 2,261 (55%) | 567 (14%) | 2,976 (72%) | 597 (14%) |
| 5 | Feb 1-7/21 | 1,800 (43%) | 2,340 (57%) | 2,295 (55%) | 591 (14%) | 3,019 (73%) | 530 (13%) |
| 6 | Feb 8-14/21 | 1,800 (43%) | 2,340 (57%) | 2,237 (54%) | 574 (14%) | 3,013 (73%) | 553 (13%) |
| 7 | Feb 15-21/21 | 1,790 (43%) | 2,340 (57%) | 2,294 (56%) | 567 (14%) | 3,013 (73%) | 550 (13%) |
| 8 | Feb 22-28/21 | 1,800 (44%) | 2,250 (56%) | 2,206 (54%) | 544 (13%) | 2,966 (73%) | 540 (13%) |
| 9 | Mar 1-7/21 | 1,800 (44%) | 2,250 (56%) | 2,221 (55%) | 543 (13%) | 2,940 (73%) | 567 (14%) |
| 10 | Mar 8-14/21 | 1,800 (44%) | 2,300 (56%) | 2,224 (54%) | 528 (13%) | 2,994 (73%) | 578 (14%) |
| 11 | Mar 15-21/21 | 1,800 (44%) | 2,330 (56%) | 2,278 (55%) | 558 (14%) | 2,982 (72%) | 590 (14%) |
| 12 | Mar 22-28/21 | 1,800 (44%) | 2,330 (56%) | 2,290 (55%) | 564 (14%) | 3,007 (73%) | 559 (14%) |
| 13 | Mar 29-Apr 4/21 | 1,790 (44%) | 2,250 (56%) | 2,224 (55%) | 555 (14%) | 2,971 (74%) | 514 (13%) |
| 16 | Apr 19-25/21 | 1,790 (44%) | 2,320 (56%) | 2,320 (56%) | 613 (15%) | 2,942 (72%) | 555 (14%) |
| 17 | Apr 26-May 2/21 | 1,800 (42%) | 2,490 (58%) | 2,345 (55%) | 611 (14%) | 3,095 (72%) | 584 (14%) |
| 18 | May 3-9/21 | 1,790 (42%) | 2,500 (58%) | 2,381 (56%) | 588 (14%) | 3,097 (72%) | 605 (14%) |
| 19 | May 10-16/21 | 1,800 (42%) | 2,500 (58%) | 2,415 (56%) | 592 (14%) | 3,076 (72%) | 632 (15%) |
| 20 | May 17-23/21 | 1,800 (42%) | 2,500 (58%) | 2,371 (55%) | 577 (13%) | 3,116 (72%) | 607 (14%) |
| 21 | May 24-30/21 | 1,800 (42%) | 2,490 (58%) | 2,367 (55%) | 580 (14%) | 3,082 (72%) | 628 (15%) |
| 22 | May 31-6/21 | 1,800 (42%) | 2,500 (58%) | 2,448 (57%) | 603 (14%) | 3,060 (71%) | 637 (15%) |
| 23 | Jun 7-13/21 | 1,800 (42%) | 2,500 (58%) | 2,428 (56%) | 599 (14%) | 3,056 (71%) | 645 (15%) |
| 24 | Jun14-20/21 | 1,800 (42%) | 2,500 (58%) | 2,420 (56%) | 587 (14%) | 3,068 (71%) | 645 (15%) |
| 25 | June 21-27/21 | 1,790 (42%) | 2,490 (58%) | 2,423 (57%) | 576 (13%) | 3,082 (72%) | 622 (15%) |
| 26 | Jun 28-Jul 4/21 | 1,800 (44%) | 2,330 (56%) | 2,322 (56%) | 569 (14%) | 2,966 (72%) | 595 (14%) |
| | **Total** | **80,340 (43%)** | **105,474 (57%)** | **103,454 (56%)** | **19,377 (11%)** | **139,716 (75%)** | **26,721 (14%)** |

*No data on weeks 14 and 15

**Proper physical distancing.** Proper physical distancing was the second NPI practice the community complied with next to respiratory hygiene. The proportion of proper physical distancing at a national level was lowest at 14% and highest above 22% in the observation weeks. Similar to respiratory hygiene, the declining weekly trend from the start showed a small increment around the 13th through the 16th weeks (Fig 2). The proportion of proper physical

distancing across the observation period in Addis Ababa with lowest at 21% and highest above 31% whereas the proportion in major regional cities was lowest at 8% and highest above 18%. There is a big difference in proper physical distance practice between Addis Ababa city and regional cities. This difference is again statistically significant at $P$ = 0.000 (Fig 3).

**Proper hand hygiene.** Proper hand hygiene was the least NPI practice the community complied with. The proportion at national level was lowest at 4% and highest above 10% during the observation weeks (Fig 2). The overall proportion of proper hand hygiene practice showed a declining trend in both Addis Ababa city and regional cities. However, there was still a big difference in the proportion of proper hand hygiene between the two population groups, ranging from 7% - 16% and 2% - 7%, respectively. This difference is also statistically significant at $P$ = 0.000 (Fig 3).

**Covid -19 deaths and public health measures.** The trends of weekly Covid—19 deaths have two peaks from August 03, 2020 to July 4, 2021; at the 34[th] of 2020 epidemiologic week 150 people died and in the 15[th] of 2021 epidemiologic week the death count rose to 211. The government has also taken several varieties of public health measures during these periods including legislation, enforcement, quarantine, passive case detection, financial package, individual measures, public awareness campaigns, no masks no service campaigns, etc. As it was depicted in Fig 2, the overall proper NPI compliance level also showed an increment during these periods.

## Variation of proper NPI practice in service facilities

**Proper respiratory hygiene.** An overall analysis combining the 39 weeks observation data showed that the proportion of proper respiratory hygiene compliance at the different service facilities at national level ranges from 25% - 55%; the highest compliance was observed at the health facility and the least at food and drink establishments. The stratified analysis of proper respiratory hygiene by cities for the different facilities ranged from 40% - 80% and 14% - 38% in Addis Ababa and regional cities, respectively. The highest public compliance was observed in Addis Ababa at all the service facilities compared to the regional cities *(P* = 0.000) (Fig 4).

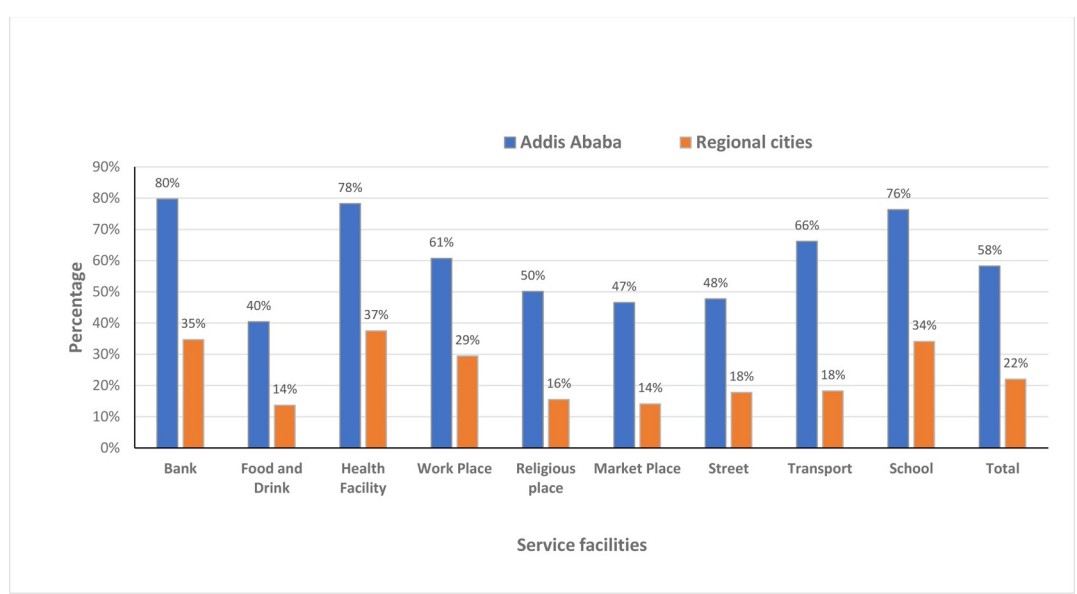

**Fig 4. Proper respiratory hygiene at different service facilities in capital Vs regional cities, Oct 5/2020-July 4/2021.**

**Proper physical distancing.** The proper physical distancing public compliance at the different service facilities at the national level ranges from 9% - 23%. Like the respiratory hygiene the highest compliance was recorded at the bank but the least at the transport service. The stratified analysis of proper physical distancing by observation places for the different facilities ranges from 13% - 35% and 7% - 16% in Addis Ababa and regional cities, respectively. The highest and the least proportion of compliance in Addis Ababa were at the bank and transport service facilities, respectively. Although the least proportion of compliance in regional cities was on the transport service, but the highest compliance was at the workplace. An overall highest public compliance was observed in Addis Ababa at all the service facilities compared to the regional cities. This difference is statistically significant at *P* = 0.000 (Fig 5).

**Proper hand hygiene.** The proper hand hygiene public compliance at the different service facilities at the national level ranges from 0.2% - 16%. The highest compliance was recorded at the food and drink establishments but the least at the transport service. The stratified analysis of proper hand hygiene by cities for the different facilities ranges from 0.3% - 20% and 0.1% - 12% in Addis Ababa and regional cities, respectively. Similar to the national level, the highest and the least proportion of compliance was recorded at the same facilities in both cities. The overall highest public compliance was observed in Addis Ababa city at all the service facilities compared to the regional cities, with a statistically significant at *P* = 0.000 (Fig 6).

## Variation of proper NPI practice by sex and age

A stratified analysis was performed to observe NPI practice difference by sex and age group within the observation city. The proper practice of the three NPIs in Addis Ababa was more than that in the regional cities, with statistical differences (*P* = 0.000). Females tend to have increased respiratory hygiene practice relative to males, while the age group greater than 50 had better physical distancing than other age groups (Table 2).

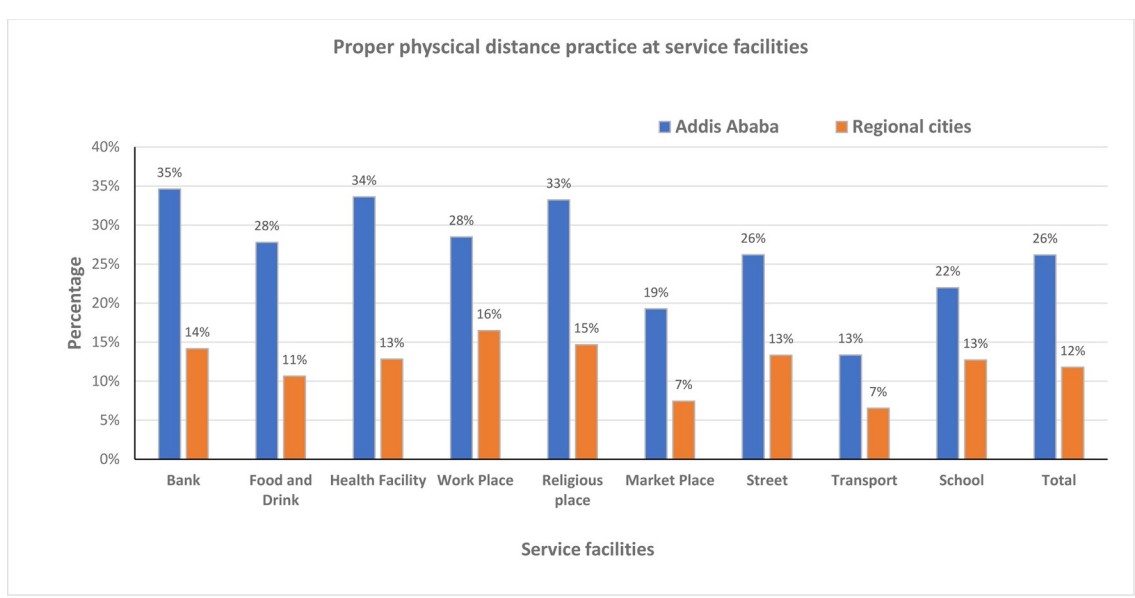

**Fig 5. The proper physical distancing at different service facilities in capital vs regional cities, Oct 5/2020-July 4/2021.**

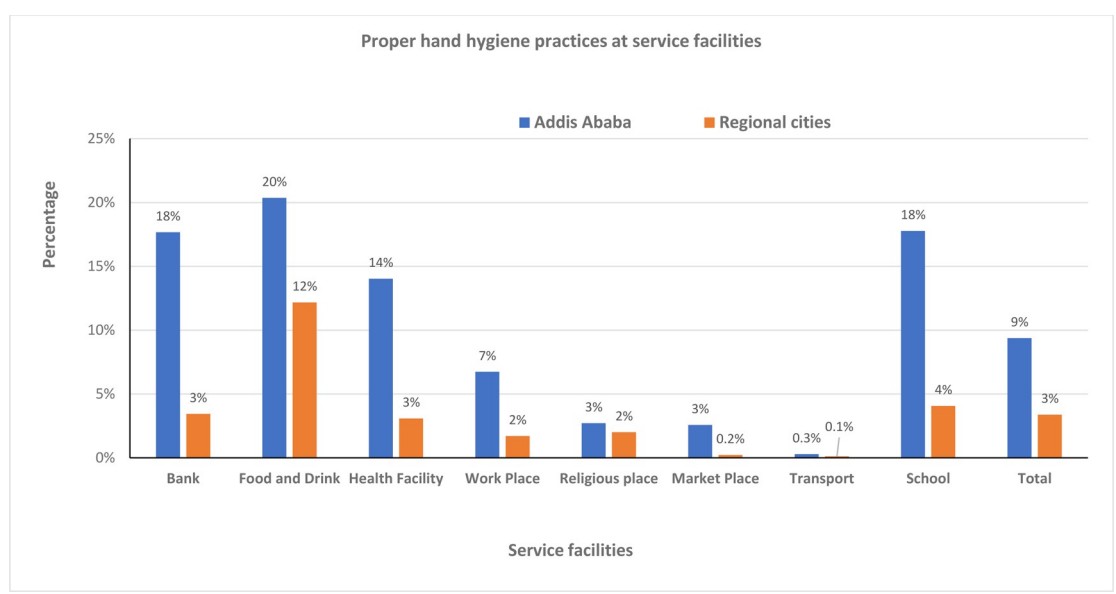

**Fig 6. Proper hand hygiene at different service facilities in capital Vs regional cities, Oct 5/2020-July 4/2021.**

## Discussion

The overall trend of NPI practice reached at its peak during the 13th– 15th weeks and declined towards the end of this study, after the 16th epidemiological week. Respiratory hygiene is the most widely practiced NPI than the other two NPIs across all study sites. Generally, there was NPI compliance level difference by residence, service institutions and sociodemographic characteristics. Residents of Addis Ababa showed better compliance with all the three NPIs than population in the regions. Besides, the highest NPI practice was observed at the formally organized service facilities such as workplaces, health facilities and banks.

The NPI compliance showed a sporadic trend with two major peaks. Following the full enforcement of the state of emergency declaration, another study in Addis Ababa reported that NPI compliance levels rises closer to peak 80% on June 2020 [2]. Three months later, the current nation-wide monitoring study started, and the NPI compliance level of 41% for respiratory hygiene was recorded, which was on the 41st epidemiological week. For the rest of the observation period, there was no significant change in the level of NPI compliance, until it reached to the peak around the 13th - 15th epidemiological weeks. Following the enforcement of the Covid-19 Prevention Directive and the "No Mask No Service" campaign initiatives by the government at the last week of March, the community compliance level to all the three

**Table 2. Proper NPI practices by sex and age group in the capital Vs regional cities, Oct 5/2020-July 4/2021.**

| NPI practices | Observation Cities | Sex | | | Age, years | | | |
|---|---|---|---|---|---|---|---|---|
| | | Male, n (%) | Female, n (%) | P-value | <18, n (%) | 18–50, n (%) | >50, n (%) | P-value |
| **Proper respiratory hygiene** | Addis Ababa | 23,264 (55%) | 23,510 (62%) | 0.000 | 5,183 (64%) | 35,586 (58%) | 6,005 (57%) | 0.000 |
| | Regional cities | 12,968 (21%) | 10,257 (23%) | 0.000 | 2,390 (21%) | 16,951 (22%) | 3,884 (24%) | 0.000 |
| **Proper physical distancing** | Addis Ababa | 11,747 (28%) | 9,276 (24%) | 0.000 | 1,620 (20%) | 16,171 (26%) | 3,232 (31%) | 0.000 |
| | Regional cities | 7,539 (12%) | 4,887 (11%) | 0.000 | 1,036 (9%) | 9,067 (12%) | 2,323 (14%) | 0.000 |
| **Proper hand hygiene** | Addis Ababa | 3,416 (8%) | 3,144 (8%) | 0.267 | 1,089 (13%) | 4,723 (8%) | 748 (7%) | 0.000 |
| | Regional cities | 2,064 (3%) | 1,030 (2%) | 0.000 | 259 (2%) | 2,433 (3%) | 402 (3%) | 0.000 |

NPIs started to rise at the 12[th] week and continued for four weeks. Our observation was interrupted for a couple of weeks during the high compliance period; hence, we couldn't report the exact peak week.

Every government world-wide took various public health measures at different time to reduce the spread of Covid-19 infection [1, 21–23]. During the 39 weeks of our observation period, a total of 59 WHO documented Covid-19 prevention measures were implemented in Ethiopia. Nine of these new interventions were implemented in March 2021. In March, the intensity of public measures was the highest (on average 2–3 new interventions per week), whereas it was on average less than 1.5 interventions per week in the rest of the months. Besides, many of the public measures were related to the mandatory face mask use in public places, followed by physical distancing. Although there were newly introduced interventions related to general public awareness, policy and regulation, during this period, there was no intervention initiative specific to hand hygiene practice; hence, this might be the reason we didn't observe changes in public practice [1].

The rising community compliance level during the epidemiological weeks of 13[th]– 15[th] might also be linked with several factors. One of the factors might be the trends of new cases and deaths due to Covid-19. The second wave of Covid-19 infection in Ethiopia was also on the rise during this period. The 13[th] and 14[th] epidemiological weeks of 2021 were the highest peak for Covid-19 new cases and new deaths, respectively, and both incidences decline after the peak weeks [12]. Very interestingly, the proper NPI compliance level also showed a similar decreasing pattern (Fig 2). An increased rate of new infection and fatalities during these periods might also play the key role to initiate more serious public measures by the government, and the public may panic with the incidence information that could led individuals to comply with the NPI measures. This study revealed that both the government initiated strong enforcement and increased incidence rate were closely linked with the increased NPI compliance. Several studies have also showed the connection between NPI measures and respiratory diseases pandemics including incidence of Covid 19 [24–27]. A large cohort in the US population indicated that NPI measures specifically public mask mandate significantly associated with Covid-19 incidence [25].

Most of NPI related studies focused on the effectiveness of different and combined NPIs on the control of transmission, but information regarding public compliance level with time was limited. In some studies, organized efforts initiated by governments, public awareness advocacy and risk communication programs affect NPI adherence in a community [28–30]. A population based bi- weekly survey (April to November 2020) from the US showed a substantial decrease of NPI adherence index from 70 at the beginning of the survey, maintain plateau at 50 during June and a slight increase to 60 in November [30]. The current authors also reported a similar change in pattern of NPI compliance [2]. Another community-based study in Nigeria that examined Covid-19 community mitigation practices found that wide ranges of government measures, healthcare policy and public campaign programs related to the pandemic affected populations behavior towards NPIs practices [29].

Throughout the observation period, proper respiratory hygiene compliance was consistently the highest, among the three NPI practices in this study. At the beginning of the pandemic, social distancing and handwashing were well accepted NPIs compared to the face mask as the most effective strategy to control the transmission, worldwide [31]. Before the current guideline, even WHO had declared as there was no evidence about face mask in protecting healthy individuals from Covid-19 infection [32]. However, the embarking message by CDC about the benefit of face covering for the prevention of Covid-19 had influenced the practice of population towards mask use [33]. Lately, the prevention of Covid-19 transmission through proper respiratory hygiene (wearing mask) is widely advocated throughout the world. On top

of the regulatory public measures, several scientific evidence were advised from recognized public health institutions promoting face covering as an efficient prevention method, particularly for resource limited settings [15, 33–35]. Accessibility of this information to the public and the increased rate of Covid-19 transmission might have influenced the decision-making power of individuals to comply to face mask than the other NPIs. The feasibility to enforce the mandatory use of face-mask, at individual level while in service facilities and public places, might also be a reason for the higher proper respiratory hygiene compliance than other NPIs in the current study [5].

The variation of compliance to each NPIs is affected by several factors. The current study highlighted the variation of NPI compliance of individuals by place of residence, socio-demographic characteristics and service provision facilities. People living in the capital city had a better compliance to the overall NPIs compared to regional cities residents. Except on physical distancing females had better compliance to the other NPIs than males and people at the workplaces, health facilities and bank services show better NPI compliance than at the transport services. Abdelhafiz and colleagues identified urban resident Egyptians had better knowledge and NPI practice compared to the rural residents [36]. Females were more likely to follow good preventive practices, comply with wearing mask and hand hygiene than men in different population-based studies in Africa [2, 29, 37]. A previous weekly monitoring of NPIs at different service facilities in Ethiopia have also reported a similar result [2]. The availability of information, restrictions and required procedures might be the main reason for the better compliance with NPI measures at institutions.

Although the weekly monitoring of NPIs practice with data from the community, covering a wide geographic area and national representation is vital for decision making. This study has also shared several limitations, which was mentioned in authors previous publication [2]. In addition, we are not able to determine the impact of localized interventions by regional governments and the pressure of global Covid-19 related interventions and information in the study. Moreover, this study only tracks people NPI practice at the selected service facilities which may not confirm their compliance in other places.

## Conclusions and recommendations

The weekly trend of community NPI compliance followed a similar trend of rising with Covid-19 incidence and the strong government-initiated public measures. Community adherence to NPI practices was not sustainable after the decline in incidence and lack of new interventions. Compliance to proper respiratory hygiene was by far higher than hand hygiene and physical distancing. Besides, the overall NPI practice level was varied by place of resident, sociodemographic characteristics of individuals and service provision facilities. Therefore, strengthening government-initiated risk communication and advocacy measures should be taken as a valid strategy to increase community compliance to NPI practices, possibly for similar pandemics in the future.

## Supporting information

**S1 Raw image. COVID-19 confirmed cases, recovery and death by Epi-Week as of July 04, 2021, Ethiopia.**
(DOCX)

**S1 File. Study protocol.**
(DOCX)

**S2 File. Data collection checklist.**
(DOCX)

**S3 File. IRB ethics approval letter.**
(PDF)

**S1 Data.**
(XLSX)

## Acknowledgments

We would like to thank the Federal Democratic Republic of Ethiopia Ministry of Health and the College of Health Sciences and the School of Public Health of Addis Ababa University.

## Author Contributions

**Conceptualization:** Yifokire Tefera, Abera Kumie, Damen Hailemariam, Samson Wakuma, Teferi Abegaz.

**Data curation:** Yifokire Tefera, Abera Kumie, Samson Wakuma, Teferi Abegaz, Shibabaw Yirsaw.

**Formal analysis:** Yifokire Tefera, Abera Kumie, Samson Wakuma, Teferi Abegaz, Mulugeta Tamire, Shibabaw Yirsaw.

**Investigation:** Yifokire Tefera, Abera Kumie, Samson Wakuma.

**Methodology:** Yifokire Tefera, Abera Kumie, Damen Hailemariam, Samson Wakuma, Teferi Abegaz, Mulugeta Tamire.

**Project administration:** Yifokire Tefera, Abera Kumie, Damen Hailemariam, Samson Wakuma, Teferi Abegaz.

**Resources:** Yifokire Tefera.

**Software:** Yifokire Tefera, Abera Kumie, Teferi Abegaz.

**Supervision:** Yifokire Tefera, Abera Kumie, Samson Wakuma, Teferi Abegaz, Mulugeta Tamire, Shibabaw Yirsaw.

**Validation:** Yifokire Tefera, Abera Kumie, Samson Wakuma.

**Visualization:** Yifokire Tefera.

**Writing – original draft:** Yifokire Tefera.

**Writing – review & editing:** Yifokire Tefera, Abera Kumie, Damen Hailemariam, Samson Wakuma, Teferi Abegaz, Mulugeta Tamire, Shibabaw Yirsaw.

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
