## [Decision Letter · Decision Letter 0]

17 May 2023

PONE-D-23-05923The impact of public-initiated COVID-19 risk communication on individual NPI practicesPLOS ONE

Dear Dr. Zele,

Thank you for submitting your manuscript to PLOS ONE. After careful consideration, we feel that it has merit but does not fully meet PLOS ONE’s publication criteria as it currently stands. Therefore, we invite you to submit a revised version of the manuscript that addresses the points raised during the review process.

We look forward to receiving your revised manuscript.

Kind regards,

Mohammed Feyisso Shaka, MPH

Academic Editor

PLOS ONE

Journal Requirements:

“We would like to thank the Federal Democratic Republic of Ethiopia Ministry of Health and the College of Health Sciences and the School of Public Health of Addis Ababa University for supporting this study.”

5. We note that Figure 1 in your submission contain map images which may be copyrighted. All PLOS content is published under the Creative Commons Attribution License (CC BY 4.0), which means that the manuscript, images, and Supporting Information files will be freely available online, and any third party is permitted to access, download, copy, distribute, and use these materials in any way, even commercially, with proper attribution. For these reasons, we cannot publish previously copyrighted maps or satellite images created using proprietary data, such as Google software (Google Maps, Street View, and Earth). For more information, see our copyright guidelines: http://journals.plos.org/plosone/s/licenses-and-copyright.

6. Please include a separate caption for each figure in your manuscript.

Reviewers' comments:

Reviewer's Responses to Questions

**Comments to the Author**

1. Is the manuscript technically sound, and do the data support the conclusions?

Reviewer #1: Partly

Reviewer #2: No

2. Has the statistical analysis been performed appropriately and rigorously? 

Reviewer #1: No

Reviewer #2: Yes

3. Have the authors made all data underlying the findings in their manuscript fully available?

Reviewer #1: Yes

Reviewer #2: Yes

4. Is the manuscript presented in an intelligible fashion and written in standard English?

Reviewer #1: Yes

Reviewer #2: No

5. Review Comments to the Author

Reviewer #1: The study aimed to assess the trend of community compliance to NPIs across cities in Ethiopia, and is important because of the number, site and geographic distribution of the observations.

Major Revisions:

INTRODUCTION

1) The article introduction states that the “Federal Ministry of Health's (FMoH) Ethiopian Public Health Institute (EPHI) takes the lead in coordination with regional partners and governments." Provide more context on the level of heterogeneity of policies and interventions across Ethiopia. Please expand the introduction to explain if regions and cities have authority to implement their own local NPI policies. Did religious locations, businesses also implement or recommend their own NPI policies that could influence the compliance at observation locations associated with their facility?

2) The introduction should provide an overview of policies, interventions, and awareness campaigns. For example, The overview of the 76 WHO interventions in the Discussion section could be moved to the introduction. Please include a brief description of the key public health policy intervention changes where the Discussion infers they have influence on NPI compliance. Examples of inference from the Discussion include the Ethiopian State of Emergency, and increased community compliance level starting in week 26. Also please provide with relevant beginning and ending dates. (Please see comment on Method on trend and temporal claims)

METHODS/CONCLUSION

3) The Methods section should be updated to include a statistical analysis of trends over time. The trend lines in Figures 1 and 2 are not sufficient to support the conclusion. For example, "Proper hand hygiene" and "Proper physical distancing" visually appear to be declining until the break in data collection, but not after the break. Also the respiratory hygiene lines in Figures 1 and 2 do not appear to decline.

4) The conclusions claim that the implementation of government initiated public measures were followed by an increased observation of NPI compliance are not support by methods. To support this claim the Methods section should be updated to include an approach to statistically test for pre-post policy and intervention changes. As noted above, the introduction should clearly explain the timing and scope of any intervention that the authors claim influenced NPI compliance.

5) Please explain the subject selection process in more detaiil. This is important because the proportion of age and gender in the population from which subjects were selected may have changed over the course of the data collection. More detail on the process to randomly select individuals can help the readers understand measurement variations over the duration of the study. For example, the proportion of observations of people less than age 18 increases over the first 17 weeks. A detailed explanation of subject selection will help the reader understand if the variation reflects a change in population from which subjects are selected or is due to the selection process itself.

6) Provide an explanation for the interrupted data collection in the 27th and 28th weeks. This interruption coincided with increased compliance observed in both Addis Ababa and regional cities around the 26-30th weeks.

RESULTS

7) Table 1, please update the manuscript to comment on the drop in observations count beginning during 17th week (Jan 25-31). For example, was change due to a change in the study or does the reduced number of observations coincide with a reduced population at the observation locations.

DISCUSSION, CONCLUSION

8) Update this section after Introduction and Methods edits to reflect the changes.

LIMITATIONS

9) Please expand on limitations. The manuscript references limitations from a prior study of much smaller scope and different time period. Examples of new limitations could include changes of population mix in the observed locations over the life of the study, unknowns of the impact of the 76 WHO interventions, and recommendations for interpreting results outside of Ethiopia.

STROBE attachment:

10) The STROBE checklist should reflect revisions to the main manuscript.

General: Please review for grammar.

Reviewer #2: 1. The title of the study and the conclusion, which focused on public initiated NPIs and their impact are not the major study findings and requires revision. The study findings were neither linked to government initiated nor public initiated NPI measures. It only presented the trends. Hence the conclusion does not reflect the findings of the study.

2. The study lacks clear definition/distinction between government initiated and public initiated NPIs use. The government started promoting NPI use for COVID 19 prevention through media and other platforms early on. Eg School closure was enforced following the first case detection in March 2020. By the time you started tracking NPIs compliance, the MOH and EPHI had been promoting NPI use through different platforms, whereby RCC was one of the strategies. Government-initiated NPI interventions are expected to be practiced and maintained by the public with good compliance to get good effect. But this does not make it public initiated.

3. It is very natural that any behavioral interventions showed high or low adherence overtime. Describing only the ups and downs using timeline have little value. Please relate the changing trend with enforced government prevention strategies, case and death detection and reporting, disease progression and burden, public panic…

4. More contextual information is required in the methods section about study sites. Eg Regional capital cities are confusing.

5. The manuscript benefit from language editing. Some comments are given on manuscript here attached.

6. PLOS authors have the option to publish the peer review history of their article (what does this mean?). If published, this will include your full peer review and any attached files.

Reviewer #1: No

Reviewer #2: No

---

## [Author Response · Author response to Decision Letter 0]

31 Jul 2023

Dear Reviewers,

In the current version of the manuscript, we tried to address all concerns and comments suggested by the reviewers. The manuscript entirely revised including the title. Major revision was made in the introduction and discussion sections. Additional literature review was included. The manuscript was language edited by two colleagues (Ansha Nega, from Queens University, Canada and Mulugeta Tamire, one of the senior authors). The details of the revision found in the uploaded revised manuscript with track change file and summarized response described in the response to reviewers file. 

Thanks

---

## [Decision Letter · Decision Letter 1]

25 Oct 2023

PONE-D-23-05923R1Impact of Covid -19 incidence rate and government-initiated risk communication measures on Individual NPI practicesPLOS ONE

Dear Dr. Zele,

Thank you for submitting your manuscript to PLOS ONE. After careful consideration, we feel that it has merit but does not fully meet PLOS ONE’s publication criteria as it currently stands. Therefore, we invite you to submit a revised version of the manuscript that addresses the points raised during the review process.

 Dear AuthorPlease find more peer reviewer report below and address the revision requested accordingly.

We look forward to receiving your revised manuscript.

Kind regards,

Mohammed Feyisso Shaka, MPH

Academic Editor

PLOS ONE

Reviewers' comments:

Reviewer's Responses to Questions

**Comments to the Author**

1. If the authors have adequately addressed your comments raised in a previous round of review and you feel that this manuscript is now acceptable for publication, you may indicate that here to bypass the “Comments to the Author” section, enter your conflict of interest statement in the “Confidential to Editor” section, and submit your "Accept" recommendation.

Reviewer #1: (No Response)

2. Is the manuscript technically sound, and do the data support the conclusions?

Reviewer #1: Partly

3. Has the statistical analysis been performed appropriately and rigorously? 

Reviewer #1: Yes

4. Have the authors made all data underlying the findings in their manuscript fully available?

Reviewer #1: Yes

5. Is the manuscript presented in an intelligible fashion and written in standard English?

Reviewer #1: Yes

6. Review Comments to the Author

Reviewer #1: Major:

1. Study aims and methods do not support showing how population NPI compliance is related to government-initiated risk communication measures. Although Introduction and Discussion do mention important government NPI actions, the Results section does not mention risk communications occurrences along the observational timeline. Also, Figures do not contain risk communication initiatives time points with the NPI compliance observations. The Figure S1_raw_image does show case and death rates, but not in relation to NPI observations. Based on results and figures provided, readers cannot link government intervention with NPI observations.

Therefore, the following content is not supported by the current methods and results:

• Title: “government-initiated risk communication measures”.

• Lines 26-27: “relation to Covid-19 incidence and government-initiated interventions”

• Line 131: “determining the trend of the weekly changes in the three individual-level NPI behaviors and its relationship with the incidence rate…”. Results and Figures do not show incident rate in separate figure, not in relation to NPI behaviors.

• Line 131: “and government-initiated measures.” Introduction and Discussion do include government initiated measures, but Results and Figures do not show government-initiated measures in relation to NPI behaviors.

• Line 390: Linkage in Figures and Results between timing of government-initiated measures should be made before justifying the following comment in the conclusion: “Community adherence to NPI practices was not sustainable after the decline in incidence and lack of new interventions.”

2. A new Results table showing beginning and end dates of the government interventions mentioned in the Introduction in relation to the “WHO Epi week *” would be helpful to future researchers that use the study data set. The study indicates there are national and many regional NPI policies. A table with policies that authors believe are most important would be sufficient.

Minor

3. Lines 46, 336, Awkward grammar, please rephrase: “compliance did not sustain longer”.

4. Line 137: In Methods, study seems to be an unobtrusive observation with repeated measures and descriptive results, not a “cross-sectional study design”. Please update study design description that more accurately describes the methods.

5. Line 170: Please clarify that the data collection tool controls the random subject selection and not the observer. It is important with many geographically scattered observers that randomization is not subject to individual observer bias.

6. Line 209, 319: Please state the reason “observation was halted for two weeks”. It is understandable that this interruption happened, but authors need to explain the reason. For example, government curfews or lock downs prevented observers from performing their tasks.

7. Line 195: Methods section mentions “appropriate hand hygiene”. File S2, Annex 1: “COVID-19 Prevention Practices Data Collection Tool in a community” lists both hand sanitization and hand washing. Please clarify if “hand hygiene” in the manuscript requires either or both to be considered “appropriate hand hygiene”.

7. PLOS authors have the option to publish the peer review history of their article (what does this mean?). If published, this will include your full peer review and any attached files.

Reviewer #1: No

---

## [Author Response · Author response to Decision Letter 1]

17 Dec 2023

Dear reviewers,

In the current submission, I filled the revised manuscript with track change indicate the level of revision and manuscript without track change that address the your comments. I also included Response to Reviewers file and the revised figures file, that includes the new recommended analysis in the result sections.

Thanks

---

## [Decision Letter · Decision Letter 2]

2 Jan 2024

PONE-D-23-05923R2Impact of Covid -19 incidence rate and government-initiated risk communication measures on Individual NPI practicesPLOS ONE

Dear Dr. Zele,

Thank you for submitting your manuscript to PLOS ONE. After careful consideration, we feel that it has merit but does not fully meet PLOS ONE’s publication criteria as it currently stands. Therefore, we invite you to submit a revised version of the manuscript that addresses the points raised during the review process.

What do the “^¶”^ and “^&”^ you put with the author’s name stands for? Please describe them if they show a contribution or other information, or remove them if not necessary. 

We look forward to receiving your revised manuscript.

Kind regards,

Mohammed Feyisso Shaka, MPH

Academic Editor

PLOS ONE

Journal Requirements:

Reviewers' comments:

Reviewer's Responses to Questions

**Comments to the Author**

1. If the authors have adequately addressed your comments raised in a previous round of review and you feel that this manuscript is now acceptable for publication, you may indicate that here to bypass the “Comments to the Author” section, enter your conflict of interest statement in the “Confidential to Editor” section, and submit your "Accept" recommendation.

Reviewer #1: All comments have been addressed

2. Is the manuscript technically sound, and do the data support the conclusions?

Reviewer #1: Yes

3. Has the statistical analysis been performed appropriately and rigorously? 

Reviewer #1: Yes

4. Have the authors made all data underlying the findings in their manuscript fully available?

Reviewer #1: Yes

5. Is the manuscript presented in an intelligible fashion and written in standard English?

Reviewer #1: Yes

6. Review Comments to the Author

Reviewer #1: I have reviewed the track changes and believe the authors have addressed the previous round of comments.

The article now incudes clearer descriptive results and sufficient background on data collection methods to allow further research using an important data set describing government policies and non-pharmaceutical intervention behavior across Ethiopia.

7. PLOS authors have the option to publish the peer review history of their article (what does this mean?). If published, this will include your full peer review and any attached files.

Reviewer #1: No

---

## [Author Response · Author response to Decision Letter 2]

27 Jan 2024

Reviewers are satisfied and approved. There is no reviewers concern.

---

## [Editor Report · Decision Letter 3]

2 Feb 2024

Impact of Covid -19 incidence rate and government-initiated risk communication measures on Individual NPI practices

PONE-D-23-05923R3

Dear Dr. Zele,

We’re pleased to inform you that your manuscript has been judged scientifically suitable for publication and will be formally accepted for publication once it meets all outstanding technical requirements.

Kind regards,

Mohammed Feyisso Shaka, MPH

Academic Editor

PLOS ONE
---

## [Editor Report · Acceptance letter]

7 Mar 2024

PONE-D-23-05923R3 

PLOS ONE

Dear Dr. Zele, 

I'm pleased to inform you that your manuscript has been deemed suitable for publication in PLOS ONE. Congratulations! Your manuscript is now being handed over to our production team.

Kind regards, 

on behalf of

Mr. Mohammed Feyisso Shaka 

Academic Editor

PLOS ONE